# Insect-Specific Chimeric Viruses Potentiated Antiviral Responses and Inhibited Pathogenic Alphavirus Growth in Mosquito Cells

Lu Tan,[a] Yiwen Zhang,[a] Dal Young Kim,[a] (ID) Runsheng Li[a,b]

[a]Department of Infectious Diseases and Public Health, Jockey Club College of Veterinary Medicine and Life Sciences, City University of Hong Kong, Hong Kong, People's Republic of China

[b]Southern Marine Science and Engineering Guangdong Laboratory (Guangzhou), Guangzhou, People's Republic of China

**ABSTRACT** Most alphaviruses are transmitted by mosquito vectors and infect a wide range of vertebrate hosts, with a few exceptions. Eilat virus (EILV) in this genus is characterized by a host range restricted to mosquitoes. Its chimeric viruses have been developed as safe and effective vaccine candidates and diagnostic tools. Here, we investigated the interactions between these insect-specific viruses (ISVs) and mosquito cells, unveiling their potential roles in determining vector competence and arbovirus transmission. By RNA sequencing, we found that these ISVs profoundly modified host cell gene expression profiles. Two EILV-based chimeras, consisting of EILV's nonstructural genes and the structural genes of Chikungunya virus (CHIKV) or Venezuelan equine encephalitis virus (VEEV), namely, EILV/CHIKV (E/C) and EILV/VEEV (E/V), induced more intensive transcriptome regulation than parental EILV and activated different antiviral mechanisms in host cells. We demonstrated that E/C robustly promoted antimicrobial peptide production and E/V strongly upregulated the RNA interference pathway components. This also highlighted the intrinsic divergences between CHIKV and VEEV, representatives of the Old World and New World alphaviruses. In contrast, EILV triggered a limited antiviral response. We further showed that initial chimera infections efficiently inhibited subsequent pathogenic alphavirus replication, especially in the case of E/V infection, which almost prevented VEEV and Sindbis virus (SINV) superinfections. Altogether our study provided valuable information on developing ISVs as biological control agents.

**IMPORTANCE** Mosquito-borne alphaviruses can cause emerging and reemerging infectious diseases, posing a considerable threat to human and animal health worldwide. However, no specific antivirals or commercial vaccines are currently available. Therefore, it is vital to develop biological control measures to contain virus transmission. Insect-specific EILV and its chimeras are supposed to induce superinfection exclusion owing to the close phylogenetical relationship with pathogenic alphaviruses. These viruses might also, like bacterial symbionts, modulate mosquito hosts' vector competence for arboviruses. However, little is known about the responses of mosquitoes or mosquito cells to ISV infections. Here, we found that EILV barely elicited antiviral defenses in host cells, while its chimeras, namely, E/C and E/V, potentiated the responses via different mechanisms. Furthermore, we showed that initial chimera infections could largely inhibit subsequent pathogenic alphavirus infections. Taken together, our study proposed insect-specific chimeras as a promising candidate for developing biological control measures against pathogenic alphaviruses.

**KEYWORDS** Eilat virus, alphaviruses, antiviral responses, chimeric viruses, mosquito cells, virus-vector interactions

Address correspondence to Dal Young Kim, paratomy89@gmail.com, or Runsheng Li, runsheng.li@cityu.edu.hk.

The authors declare no conflict of interest.

*A*lphavirus, the sole genus in the *Togaviridae* family, is a group of enveloped viruses encompassing a positive-strand genomic RNA (1). They are classified as the Old World (OW) and New World (NW) types according to where they circulate (2). Most alphaviruses are mosquito-borne and can infect a wide range of vertebrates, which renders them maintained in enzootic

vertebrate host and vector transmission cycles. Occasional spillover into human populations can result in severe outcomes. Chikungunya virus (CHIKV), a typical OW virus, can cause fever and debilitating joint pain despite low fatality. It has brought several epidemic outbreaks since 2004 (3). Venezuelan equine encephalitis virus (VEEV), one of the essential NW alphaviruses affecting humans and equines, can cause fatal encephalitis (4). Other alphaviruses, such as O'nyong'nyong virus (ONNV), Sindbis virus (SINV), Mayaro virus (MAYV), Western equine encephalitis virus (WEEV), and Eastern equine encephalitis virus (EEEV), also pose potential threats to human and animal health worldwide (5, 6). However, no specific antivirals or commercial vaccines are currently available to protect against alphavirus infections. Meanwhile, increased urbanization and climate change have expanded the distribution of many mosquito species, exaggerating virus transmission and outbreak risks (7). Therefore, it is of great importance to develop mosquito control measures. Understanding the basic biology of virus-vector interactions is a pivotal component.

In contrast to arthropod-borne alphaviruses, Eilat virus (EILV) in the same genus was characterized by an inability to infect vertebrates due to limitations at both entry and RNA synthesis levels (8, 9). Three more insect-specific alphaviruses were reported afterward, namely, Tai Forest alphavirus (TALV) (10), Mwinilunga alphavirus (MWAV) (11), and Agua Salud alphavirus (ASALV) (1). Moreover, several EILV-based chimeric viruses, consisting of the nonstructural genes of EILV and the structural genes of other alphaviruses, inherited a mosquito-only host range and have been developed as safe and effective vaccine candidates and diagnostic tools (12–14). These viruses are phylogenetically closely related to pathogenic alphaviruses. This implicates their potential to induce superinfection exclusion via competition for cellular factors or other documented mechanisms (15). Like bacterial symbionts, insect-specific viruses (ISVs) might also alter mosquito hosts' immune systems (16, 17). Taken together, mosquito-only alphaviruses can be a useful tool to modulate vector competence and contain pathogenic alphavirus transmission.

However, a critical knowledge gap remains to be filled. Little is known about the responses of mosquitoes or mosquito cells to ISV infections. Instead, virus-vector interplays have been described for several medically important arboviruses, such as CHIKV (18–22) and members of *flavivirus*, including Dengue virus 2 (DENV2) and Zika virus (ZIKV) (23–27). Despite the inability to eradicate invading viruses, mosquito vectors utilize several conserved pathways for systemic antiviral defense (28, 29). RNA interference (RNAi) is the primary mechanism and functions by degrading viral RNA or regulating host gene expression (30–33). In addition, several inducible signaling cascades also play crucial roles in antiviral responses (34–36). Activation of these signaling pathways initiates the transcription of downstream effector molecules, such as various antimicrobial peptides (AMPs), and modulates the production of innate immune regulatory factors (37). Moreover, cell death is usually dedicatedly regulated in infected cells and might play an antiviral role in some cases (38).

Although these mechanisms are conserved, the composition and intensity of induced antiviral responses can vary and have not been characterized in the context of ISVs. In a previous study, we constructed a group of EILV-based chimeras containing the structural genes derived from seven different OW and NW pathogenic alphaviruses. These chimeras displayed infection phenotypes distinct from parental EILV in *Aedes albopictus* C7/10 cells (39), suggesting potential divergences in virus-vector interactions. Also, as these chimeras consisted of structural genes of different types, they might evoke diverse defense mechanisms. The details remain unknown.

Here, we comprehensively investigated mosquito cellular responses against EILV, EILV/CHIKV (E/C), and EILV/VEEV (E/V) infections. E/C and E/V contained CHIKV's and VEEV's structural genes, respectively. Our study featured a limited antiviral response elicited by EILV and demonstrated that its chimeras could boost cellular defense by different mechanisms. E/C preferentially induced AMP production, while E/V activated the RNAi pathways. We further showed that E/C and E/V largely inhibited subsequent pathogenic alphavirus infections. Therefore, EILV-based chimeras can be deployed as a promising tool to reduce pathogenic alphavirus transmission in enzootic circulations.

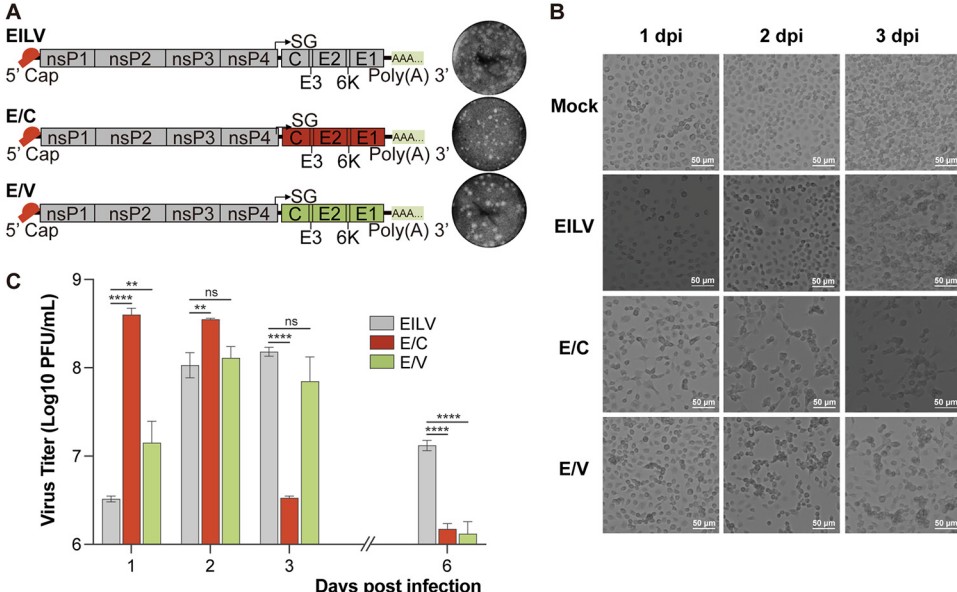

**FIG 1** Eilat virus (EILV)-based chimeras displayed distinguishable infection phenotypes from EILV in *Aedes albopictus* C6/36 cells. (A) Genome organization of EILV and EILV-based chimeras. The chimeras were generated by replacing EILV's structural genes with those of Chikungunya virus (CHIKV) or Venezuelan equine encephalitis virus (VEEV), labeled as EILV/CHIKV (E/C) and EILV/VEEV (E/V), respectively. Plaques formed on C6/36 monolayer were displayed. (B) Morphology of virus-infected C6/36 cells. Cells were infected at a multiplicity of infection (MOI) of 0.1. Phase-contrast photos were taken at 1, 2, and 3 days postinfection (dpi). (C) Growth kinetics of EILV and EILV-based chimeras. C6/36 cells were infected at MOI 0.1. Cell supernatants were collected at intended time points, and titers were determined by plaque assay. Each sample was titrated twice ($n = 2$). Average titers $\pm$ SD (error bars) are shown. * indicates differences that are statistically significant (*, $P < 0.05$; **, $P < 0.01$; ****, $P < 0.0001$).

## RESULTS

**EILV- and EILV-based chimeras reprogrammed host cell gene expression profiles.**
We previously characterized EILV-based chimeras in *Aedes albopictus* C7/10 cells (39). Here, we further described E/C's and E/V's infection phenotypes in *Aedes albopictus* C6/36 cells, one of the most widely used insect cell lines. E/C and E/V were generated by replacing the structural genes of EILV with that of CHIKV strain Caribbean or VEEV strain TC83 (Fig. 1A). Like in C7/10 cells, the chimeras formed clearer plaques than parental EILV on C6/36 monolayer (Fig. 1A). They also induced apparent cytopathic effect (CPE) and inhibited host cell multiplication, while EILV had little impact on cell growth (Fig. 1B). Consistent with the observation in C7/10 cells, the chimeras' growth curves dropped dramatically after peaking (Fig. 1C). At 6 dpi, E/C's titer showed a 268-fold decrease relative to its peak value, and E/V's titer displayed a 98-fold decrease. Instead, a modest 12-fold drop was observed in the case of EILV. These differences in infection phenotype between EILV and its chimeras suggested their variations in virus-vector interplays.

C6/36 cells were infected with EILV, E/C, or E/V, and their transcriptomes were subjected to RNA sequencing (RNA-seq) to characterize the interactions between these viruses and mosquito cells. Among generated reads, virus-specific reads accounted for a considerable portion. The relative amounts of viral reads in E/C- and E/V-infected cells were approximately twice that in EILV-infected cells (Table S1 [https://github.com/lrslab/Interplay-between-ISVs-and-mosquito-cells-suppl] and Fig. 2A). Residual reads were mapped to the C6/36 genome (40). Numerous differentially expressed genes (DEGs) were identified (Fig. 2B). Taking Fold Change (FC)>2 and adjusted $P < 0.05$ as the threshold, 1,518 DEGs were recognized in EILV-infected cells. More intensive transcriptome regulation was found by chimera infections, with 3,568 genes associated with E/C and 3640 genes related to E/V. Comparison between these DEGs revealed a substantial overlapping in transcriptome responses upon different viral infections (Fig. 2C). Of EILV-induced DEGs, 659 genes (43%) were similarly modulated during E/C infection, with 504 upregulated and 155 downregulated. A total of 1,089 communal DEGs

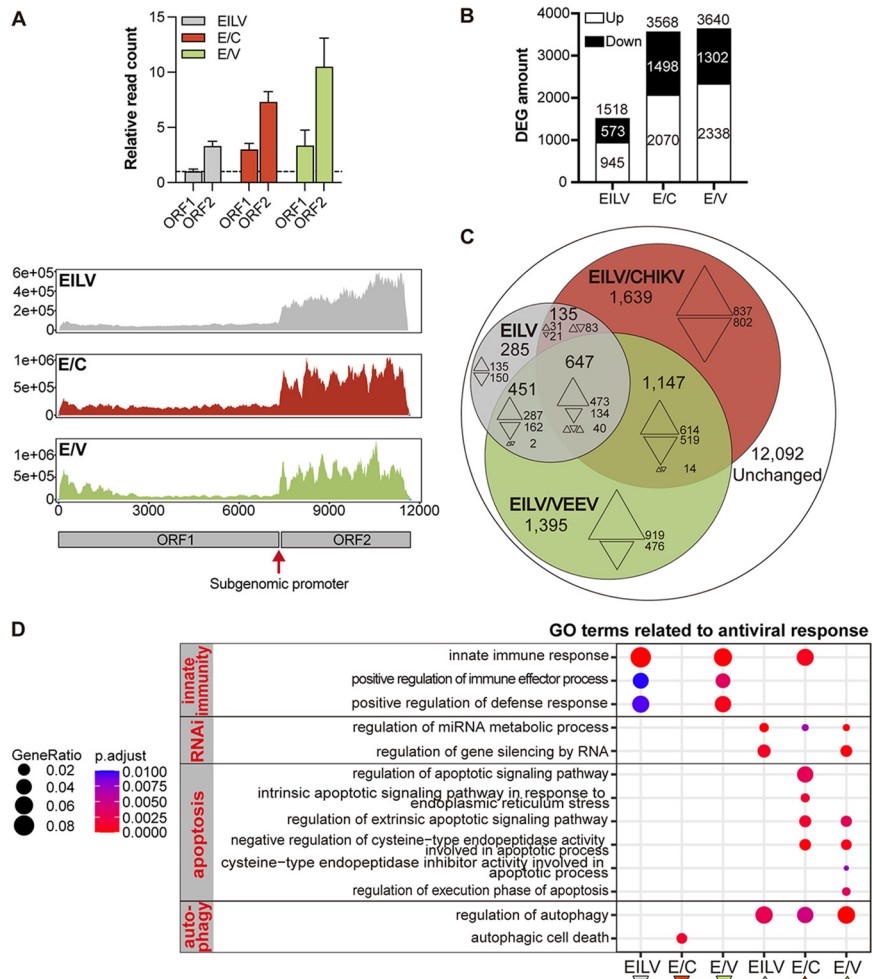

**FIG 2** EILV and its chimeras modulated host cell gene expression profiles. (A) Relative quantification of virus-specific reads and read distribution along viral genomes. Read counts mapped to the open reading frame (ORF) 1 and 2 were normalized to that of the 60S ribosomal protein L23 gene (RPL23). Mean ± SD are shown; n = 3. (B) Summary of differentially expressed genes (DEGs) in host cells. Cellular transcriptomes were mapped to C6/36 genome using Tophat2, and DEGs were identified with egdeR. Fold Change (FC)>2 and adjusted $P < 0.05$ were taken as a threshold. (C) Venn diagram showing the relationship of DEGs induced by different viral infections. (D) Gene ontology (GO) terms related to antiviral responses. GO enrichment analysis was performed by clusterProfiler. GO terms related to innate immunity, RNA interference, apoptosis, and autophagy were selectively displayed.

(72% of EILV-induced DEGs) were detected with EILV and E/V infections, of which 761 were upregulated and 328 were downregulated. However, despite the crossover, 1,639 DEGs were specific for E/C infection, and 1,395 DEGs were specific for E/V infection, indicating significant variations between these viruses.

Gene ontology (GO) analysis showed that viral infections profoundly changed the host cells' biological activities (Fig. S1 [https://github.com/lrslab/Interplay-between-ISVs-and -mosquito-cells-suppl] and 2D; Table S2 [https://github.com/lrslab/Interplay-between-ISVs -and-mosquito-cells-suppl]). Among the top enriched GO terms, a group of genes downregulated by chimera infections were recognized as related to DNA replication and cell cycle (Fig. S1 [https://github.com/lrslab/Interplay-between-ISVs-and-mosquito-cells-suppl]). This might explain the arrested cell multiplication observed in chimera-infected cells (Fig. 1B). Moreover, viral infections significantly altered host antiviral responses (Fig. 2D). EILV and E/V infections commonly downregulated innate immune response-related genes. In contrast, E/C infection significantly upregulated genes in the same category. EILV and E/V infections enhanced the regulation of gene silencing pathways. In terms of cell death, chimera infections broadly modulated host cell apoptotic cascade, either in a

**TABLE 1** Regulation of immune signaling pathways by EILV, E/C, and E/V via RNA-seq analysis[a]

| Pathway | Symbol | Gene ID | EILV | E/C | E/V |
|---------|--------|---------|------|-----|-----|
| Toll pathway | GNBPA1 | LOC109407966 | 0.63 | **2.03** | 0.82 |
|  | Spz | LOC109424643 | 0.59 | 0.95 | **0.46** |
|  | Spz | LOC109412941 | 0.74 | 0.85 | **0.47** |
|  | Spz | LOC109424625 | **2.10** | **0.38** | 1.61 |
|  | Tl | LOC109400880 | 1.60 | 1.31 | **2.38** |
|  | Tl | LOC109402620 | 1.54 | **0.17** | **0.24** |
|  | Tl | LOC109406617 | 1.25 | **0.14** | **0.44** |
|  | Tehao | LOC109402615 | **0.47** | 0.88 | 0.82 |
|  | TOLL2 | LOC109419298 | 1.43 | **2.83** | **3.92** |
|  | TOLL2 | LOC109426104 | 1.14 | 1.60 | **2.27** |
|  | TOLL7 | LOC109403397 | 1.08 | **5.86** | **4.06** |
|  | TOLL7 | LOC109402490 | 1.41 | **6.23** | **2.66** |
|  | TOLL10 | LOC109402284 | 1.84 | **0.22** | **2.58** |
|  | TOLL10 | LOC109420429 | 1.10 | **0.12** | 1.51 |
|  | TOLL11 | LOC109402341 | 1.06 | **0.18** | 0.36 |
|  | TOLL11 | LOC109420424 | 0.73 | **0.11** | 0.36 |
|  | Cact | LOC109429544 | 0.67 | 1.00 | **0.45** |
| IMD pathway | PGRPLB | LOC109422736 | **0.24** | 0.74 | **0.23** |
|  | PGRPLB | LOC109431862 | **0.23** | 1.58 | **0.27** |
|  | Fadd | LOC109428687 | 0.65 | 0.88 | **0.50** |
|  | Dredd | LOC109401234 | 1.24 | 0.64 | **2.16** |
|  | IAP2 | LOC109402332 | 0.97 | 1.35 | **2.39** |
|  | IAP2 | LOC109420403 | 1.02 | 1.27 | **2.20** |
|  | Rel2 | LOC109403656 | 1.04 | **2.66** | 1.64 |
|  | Rel2 | LOC109407961 | 0.93 | **3.07** | 1.59 |
| JAK-STAT pathway | Hop | LOC109420726 | 0.93 | **0.28** | 0.55 |
|  | Hop | LOC109419906 | 0.96 | **0.30** | 0.51 |
|  | Socs36E | LOC109425964 | 1.36 | **2.46** | **2.66** |
|  | Socs36E | LOC109407638 | 1.24 | **2.25** | **2.53** |

[a]FC values greater than 2 with adjusted $P < 0.05$ are highlighted in bold.

pro- or anti-apoptotic way; both EILV and chimeras significantly upregulated genes regarding autophagy regulation. Above mentioned antiviral responses play an irreplaceable role in inhibiting virus growth and modifying vector competence. More details will be discussed below.

**E/C efficiently induced AMP production.** Although there is a considerable crossover, the DEGs induced by EILV, E/C, and E/V indicated great divergences between these viruses in initiating and modifying antiviral responses in host cells (Fig. 2D). According to RNA-seq analysis, chimera infections significantly regulated the transcription of some immune signaling pathway components (Table 1), including those in the Toll (34), immune deficiency (IMD) (35), and Janus kinase/signal transducers and activators of transcription (JAK-STAT) pathways (36). Both positive and negative factors were up- or downregulated, complicating the regulatory network. In contrast to chimera infections, EILV infection modulated few of these signaling components (Table 1).

Activation of immune signaling pathways leads to the generation of various effector molecules. Transcriptional activators Rel1 and Rel2 in the Toll and IMD pathways are key regulators of AMP production (Fig. 3A) (37, 40). AMPs are a group of peptides equipped with antibacterial and antiprotozoal activity and show great potential in inhibiting viral infections (41). Several genes were identified in the C6/36 genome putatively encoding six AMPs (Table S3 [https://github.com/lrslab/Interplay-between-ISVs-and-mosquito-cells-suppl]). These are Attacin B (AttB), Attacin C (AttC) (42), Defensin A (DEFA), Defensin B (DEFB), Defensin C (DEFC) (43), and C-type lysozyme (LYSC11) (44, 45). Despite complex modulation of the Toll and IMD pathway components (Table 1), AMP production induced by infecting viruses was of regularity, which was validated by quantitative reverse transcription-PCR (RT-qPCR) analysis (Fig. 3B). It is seen that E/C constantly potentiated the transcription of four out of six putative AMP products, including AttB, DEFA, DEFB, and DEFC. DEFA was most intensively upregulated with

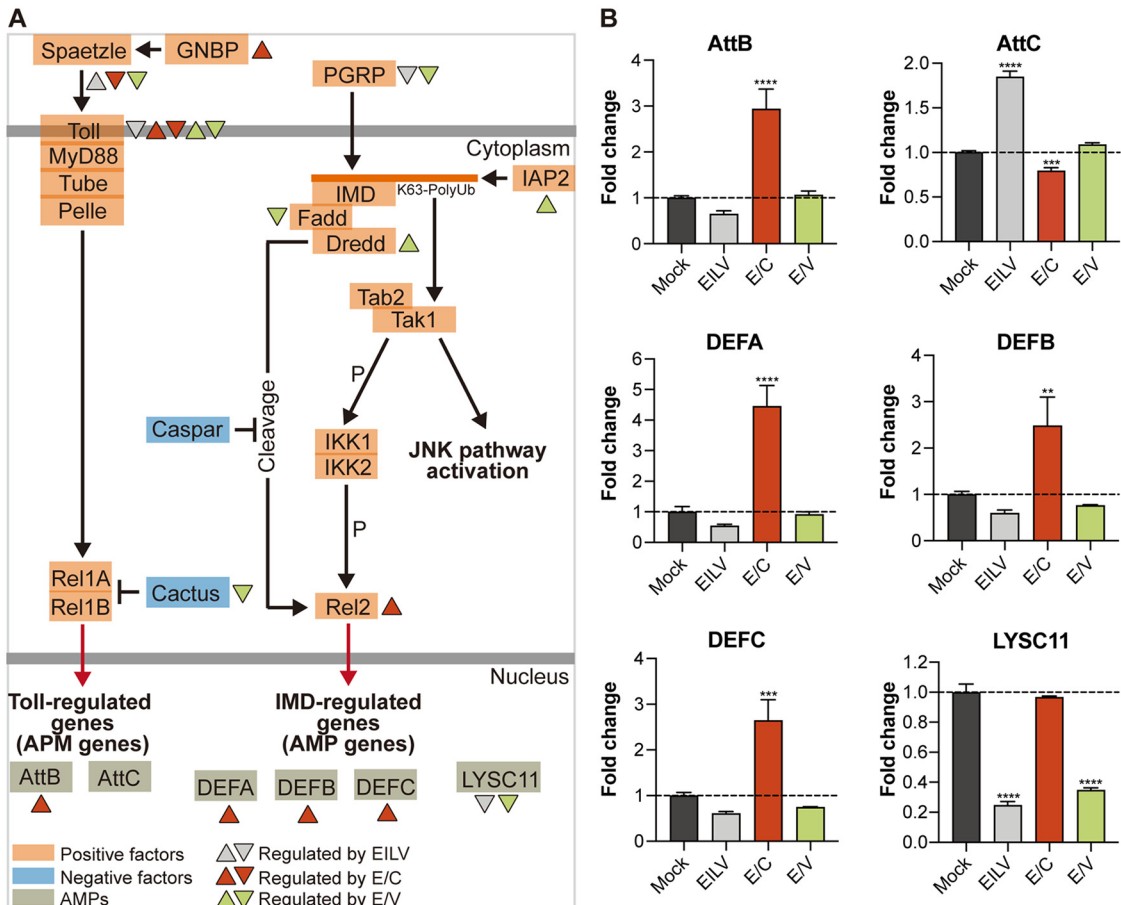

**FIG 3** E/C significantly induced the production of antimicrobial peptides (AMPs). (A) Schematic of mosquito Toll and immune deficiency (IMD) pathways, the key regulators in AMP production, modified from (34, 72, 73). Regulation of signaling pathway components was indicated according to RNA-seq analysis (FC > 2 and adjusted $P$ < 0.05). Regulation of AMP production was indicated based on quantitative reverse transcription-PCR (RT-qPCR) analysis (FC > 2 and $P$ < 0.05). (B) Fold changes in the expression of AMP genes following viral infections. Expression levels of Attacin B (AttB), Attacin C (AttC), Defensin A (DEFA), Defensin B (DEFB), Defensin C (DEFC), and C-type lyzosyme (LYSC11) were determined via RT-qPCR. Mean ± SD are shown; $n$ = 3. The dashed line indicates an FC value of 1, which corresponds to an unchanged gene expression level. * indicates differences that are statistically significant between virus- and mock-infected groups (**, $P$ < 0.01; ***, $P$ < 0.001; ****, $P$ < 0.0001).

a fold change greater than four. The others increased to at least 2-fold. On the contrary, AMP expression levels remained low in EILV- and E/V-infected cells. Notably, EILV and E/V infections significantly reduced the transcription of LYSC11 by more than half, whereas E/C infection did not affect its production.

**E/V robustly activated RNAi pathways.** In addition to inducible immune signaling pathways, RNAi is another key antiviral mechanism in mosquitoes, including the small interfering RNA (siRNA) (31), micro-RNA (miRNA) (32), and P-element-induced wimpy testis (PIWI)-interaction RNA (piRNA) pathways (33, 46) (Fig. 4A). They were significantly modulated by viral infections (Fig. 2D). In the siRNA pathway, Dicer-2 (Dcr-2) is the most crucial protein that cleaves the viral dsRNA intermediates to produce siRNAs. Here, it is shown that E/V infection strongly boosted Dcr-2 transcription to approximately 17-fold. E/C induced Dcr-2 production to about 4-fold, while EILV had little effect on it (Fig. 4B). However, C6/36 cells used in the present study are thought to have an impaired siRNA response, which results from a lack of dicing activity of Dcr-2 due to a homozygous frameshift mutation (47–49). Therefore, robust Dcr-2 upregulation in E/V-infected C6/36 cells might indicate a tendency to activate the siRNA pathway but will not result in an enhanced response. Protein argonaute-2 (AGO2) and double-stranded RNA-binding protein (r2d2) in the same pathway were relatively steadily transcribed (Fig. 4B).

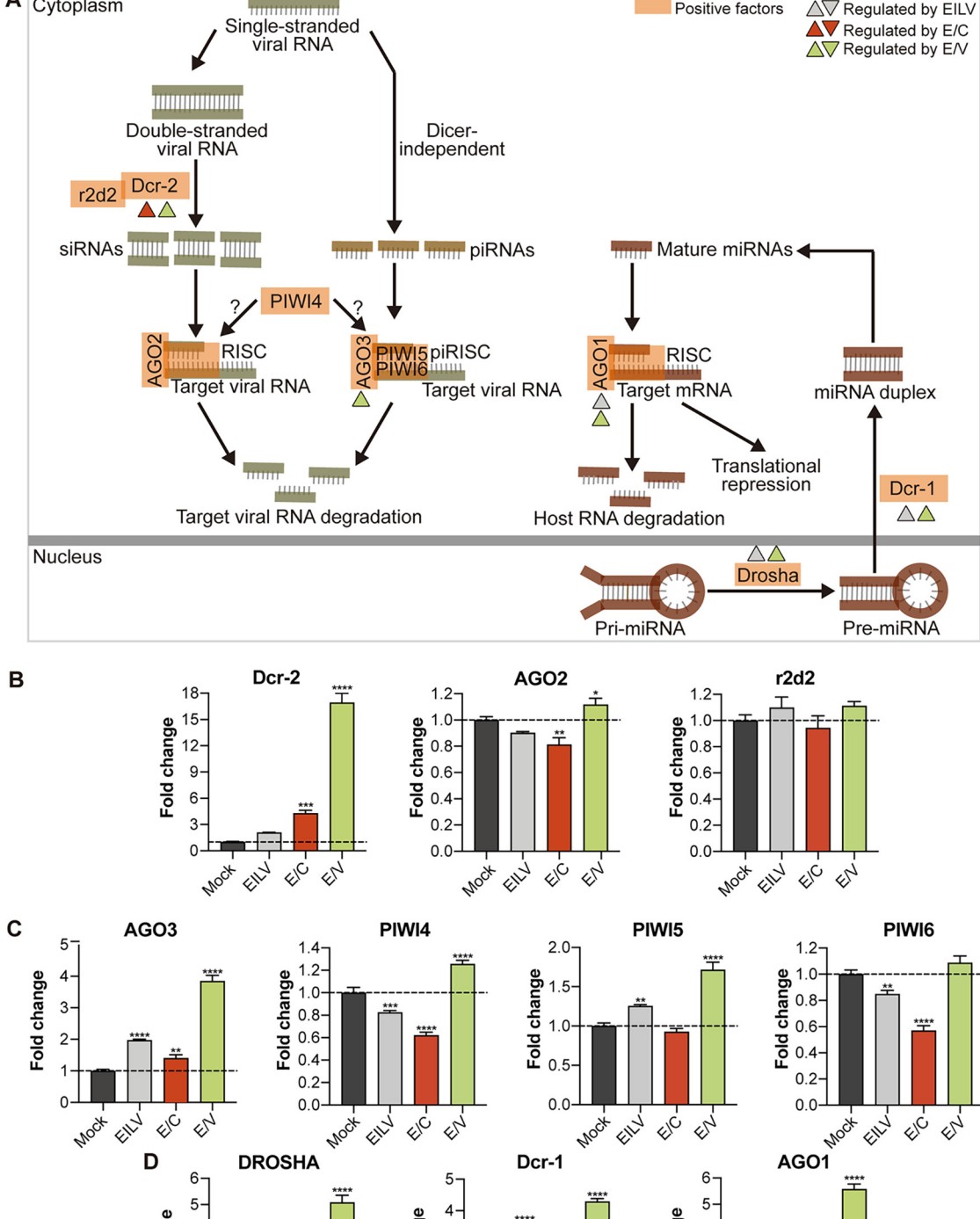

**FIG 4** E/V robustly activated the RNA interference (RNAi) pathways. (A) Schematic of mosquito RNAi pathways, modified from (37, 46, 74). Regulation of RNAi pathway components was indicated based on RT-qPCR analysis (FC > 2 and $P < 0.05$). (B) Fold changes in the expression of small interfering RNA (siRNA) pathway components after viral infections. Expression levels of Dicer-2 (Dcr-2), Argonaute 2 (AGO2), and double-stranded RNA-binding protein (r2d2) were determined via RT-qPCR. (C) Fold changes in the expression of P-element-induced wimpy testis (PIWI)-interaction RNA (piRNA) pathway components after viral infections.

Despite a dysfunctional dicing activity of Dcr-2, C6/36 cells can mount RNAi response by generating virus-derived piRNAs (50). A couple of pivotal components in piRNA biogenesis were significantly upregulated following E/V infection (Fig. 4C). Argonaute-3 (AGO3), PIWI5, and PIWI4 expressions increased to 3.85, 1.72, and 1.26-fold, respectively. Instead, E/C infection inhibited PIWI4 and PIWI6 productions to approximately 0.6-fold, regardless of a slight increase in AGO3 production. EILV infection also reduced PIWI4 and PIWI6 transcription, while upregulating AGO3 and PIWI5 expression, but to a lesser extent.

Unlike siRNAs and piRNAs degrading viral RNAs, miRNAs primarily regulate host gene expression posttranscriptionally, and a few studies have shown modified host miRNA profiles following viral infections (51, 52). The binding of specific host miRNA to the 3′ untranslated region of viral genomic RNA has also been observed, thereby limiting virus propagation in particular cell types (53). In the present study, E/V infection robustly enhanced miRNA biogenesis (Fig. 4D). Drosha, Dicer-1 (Dcr-1), and argonaute-1 (AGO-1) were significantly upregulated to 4 to 6-fold. EILV also improved the expression of these components but to a lesser extent. On the contrary, E/C posed little effect on this pathway. Although E/V and EILV infection promoted miRNA biogenesis, modulation of miRNA profiles postinfection requires further investigation.

**Viral infections modulated cell death pathways.** Many cells undergo apoptosis upon infections, which can be a crucial mechanism deployed by the host to curtail virus propagation and dissemination (54). Here, a considerable number of DEGs induced by viral infections were identified to regulate apoptotic signaling and functioning in both pro- and anti-apoptotic ways (Fig. 2D). The core apoptotic pathway in mosquitoes largely resembles that of *Drosophila melanogaster* (55, 56) (Fig. 5A). Following infections, the transcriptions of several inhibitor-of-apoptosis (IAP) genes, including IAP1, IAP2, and IAP6, were significantly potentiated, and E/V posed the most substantial effect (Fig. 5B). Meanwhile, viral infections also raised the expression levels of initiator caspase Dronc and a few effector caspases (Fig. 5A and C); fold changes triggered by E/V infection were also the greatest. To identify the apoptotic status of infected cells, terminal deoxynucleotidyl transferase dUTP nick end labeling (TUNEL) assay was performed to detect DNA fragmentation (Fig. 5D). Compared to mock-infected cells, viral infections slightly increased the number of apoptotic cells, with no significant difference between EILV and chimera infections. These results suggest that apoptosis plays a minor role in virus-vector interactions due to simultaneous pro- and anti-apoptotic regulations. Autophagy was also affected by viral infections (Fig. 2D; Table S4 [https://github.com/lrslab/Interplay-between-ISVs-and-mosquito-cells-suppl]). However, the function of autophagy in mosquitoes during arbovirus infection is understudied (57), and a limited role of autophagy has been reported (58).

**Nonstructural genes antagonized structural genes' antiviral mechanisms.** Given that E/C triggered AMP production and E/V potentiated RNAi response, it is of interest to investigate whether the corresponding parental CHIKV and VEEV induce identical antiviral responses as their chimeric constructs. To this end, total RNAs of CHIKV and VEEV-infected C6/36 cells were isolated at the time points when the viruses replicated to the highest titers (Fig. S2 [https://github.com/lrslab/Interplay-between-ISVs-and-mosquito-cells-suppl]), followed by RT-qPCR. It is shown that the expression levels of AttB, DEFA, DEFB, and DEFC were significantly higher in E/C-infected cells than in CHIKV-infected cells (Fig. 6A). CHIKV infection most robustly enhanced DEFA expression, showing a modest 2-fold increase. In contrast, DEFA transcription was upregulated to more than 4-fold in the E/C-infected cells. Moreover, like EILV and E/V infections, CHIKV infection decreased LYSC11 production by more than half. In the comparison of VEEV and E/V, despite the enhanced transcriptions of AGO3, Drosha, Dcr-1, and AGO1, the corresponding fold change values after VEEV infection were significantly lower

**FIG 4** Legend (Continued)

Expression levels of Argonaute 3 (AGO3), PIWI4, PIWI5, and PIWI6 were determined via RT-qPCR. (D) Fold changes in the expression of micro-RNA (miRNA) pathway components after viral infections. Expression levels of Drosha, Dicer-1 (Dcr-1), and Argonaute 1 (AGO1) were determined via RT-qPCR. Mean $\pm$ SD are shown; $n$ = 3. The dashed line indicates an FC value of 1, which corresponds to an unchanged gene expression level. * indicates differences that are statistically significant between virus- and mock-infected groups (*, $P < 0.05$; **, $P < 0.01$; ***, $P < 0.001$; ****, $P < 0.0001$).

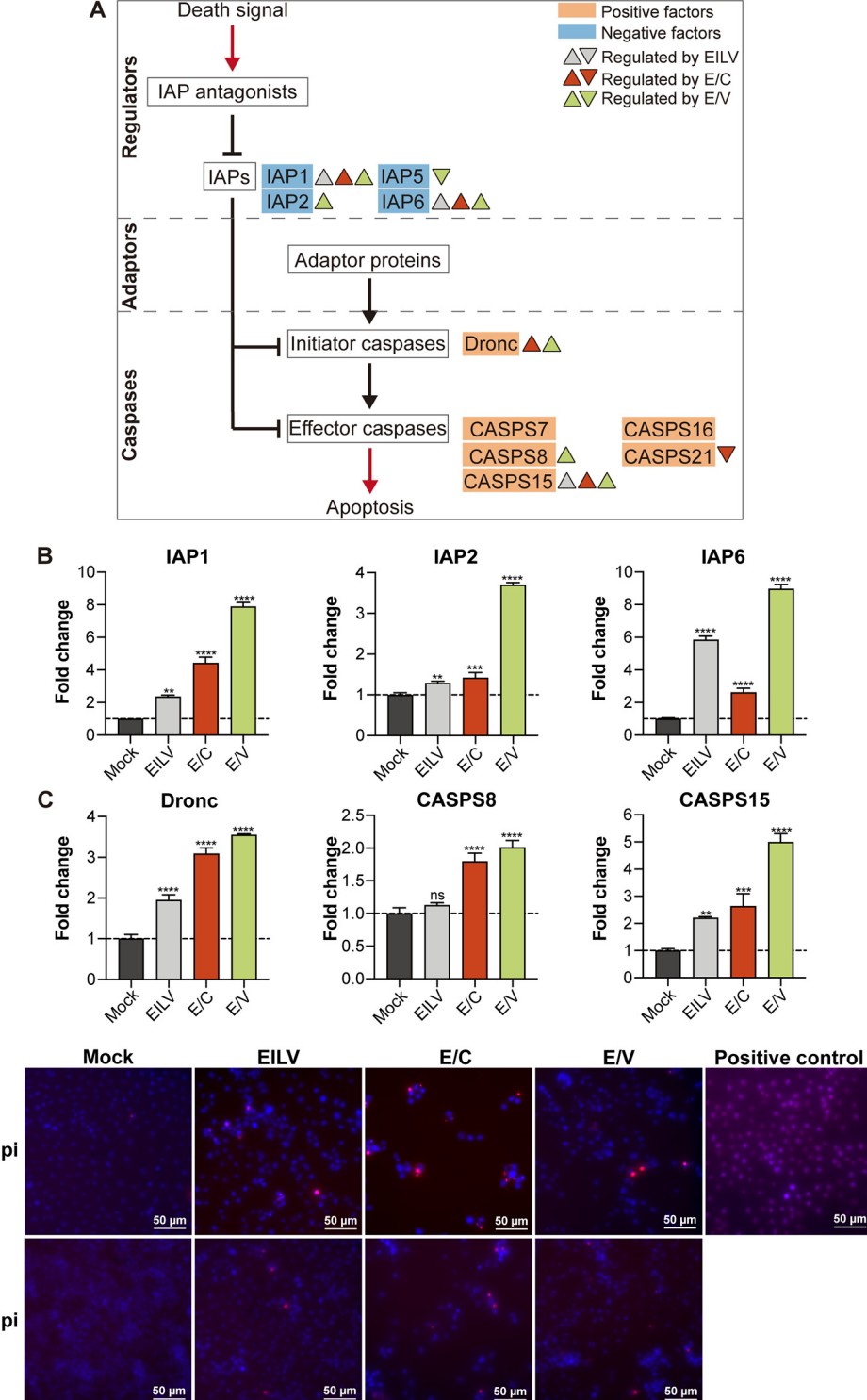

FIG 5 Viral infections modulated apoptotic response in host cells. (A) Schematic of the core apoptotic pathway in mosquitoes, adjusted from (55, 75). Regulation of pathway components was indicated according to RNA-seq (FC > 2 and adjusted $P < 0.05$) and RT-qPCR analysis (FC > 2 and $P < 0.05$). (B) Fold changes in the expression of anti-apoptotic genes. Expression levels of inhibitor-of-apoptosis (IAP) 1, IAP2, and IAP6 were determined via RT-qPCR. (C) Fold changes in the expression of proapoptotic genes. Expression levels of initiator caspase Dronc and effector caspase CASPS8 and CASPS15 were determined via RT-qPCR. Mean ± SD are shown; $n = 3$. The dashed line indicates an FC value of 1, which corresponds to an unchanged gene expression level. * indicates differences that are statistically significant between virus- and mock-infected groups (**, $P < 0.01$; ***, $P < 0.001$; ****, $P < 0.0001$). (D) Terminal deoxynucleotidyl transferase dUTP nick end labeling (TUNEL) assay of virus-infected cells. C6/36 cells were infected at MOI 0.1 and fixed at 1, 2, and 3 dpi. Positive control was treated with DNase I. Apoptotic cells were detected via TUNEL staining (red), and nuclei were labeled using DAPI (blue). The merged pictures were displayed.

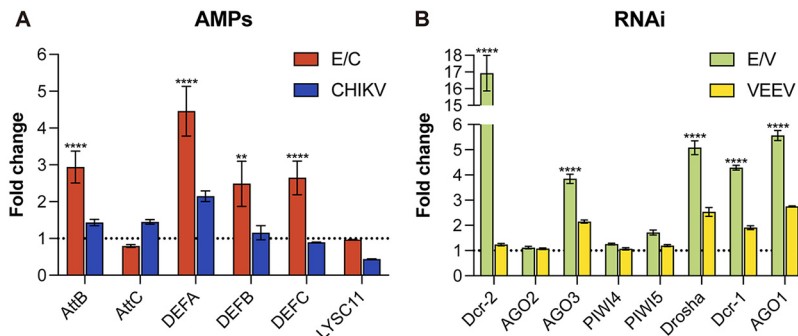

**FIG 6** Nonstructural genes antagonized structural genes' antiviral mechanisms. (A) Fold changes in the expression of AMPs following E/C and CHIKV infection. (B) Fold changes in the expression of the RNAi pathway components following E/V and VEEV infection. Expression levels of genes were determined via RT-qPCR. Mean $\pm$ SD are shown; $n = 3$. The dashed line indicates an FC value of 1, which corresponds to an unchanged gene expression level. * indicates differences that are statistically significant between chimeric and parental viruses (**, $P < 0.01$; ****, $P < 0.0001$).

than those induced by E/V infection (Fig. 6B). Notably, in contrast to the 17-fold upregulation of Dcr-2 production by E/V infection, VEEV infection did not strengthen Dcr-2 transcription. Taken together, the incorporation of parental nonstructural genes weakened the antiviral responses elicited by the structural genes, suggesting that different components of wild-type viruses coevolved to evade host defense.

**EILV-based chimeras robustly inhibited pathogenic alphavirus growth in mosquito cells.** In a previous study, EILV was proven to induce homologous and heterologous interference in *Aedes albopictus* C7/10 cells and *Aedes aegypti* mosquitoes (59). Here, the capacities of EILV-based chimeras to inhibit pathogenic alphavirus infections were investigated in C6/36 cells. EILV, E/C, or E/V were added to cell monolayers at a multiplicity of infection (MOI) of 0.1 PFU/cell to establish the initial infections. Subsequently, three pathogenic viruses were administered when the first viruses reached the highest titers, including VEEV strain TC83 containing an extra subgenomic promoter expressing green fluorescent protein (GFP), CHIKV strain 181/25, and SINV strain Toto1101. The replication of VEEV-GEP was monitored by observing GFP expression (Fig. 7A). At 4 days postsuperinfection (dpsi), all mock and EILV-infected cells displayed green fluorescence. In E/C-infected cells, GFP fluorescence was not observed until 5 dpsi. More interestingly, the initial E/V infection completely prevented GFP production. Quantitative assessment was performed by plotting the growth curves of superinfecting CHIKV, VEEV-GFP, and SINV (Fig. 7B). EILV was shown to efficiently inhibit the multiplication of second viruses at early time points. The chimeras induced more robust and durable growth interference. As E/C and E/V contain CHIKV's and VEEV's structural proteins, respectively, they profoundly inhibited the growth of homologous viruses. In the case of E/C infection, the titer of superinfecting CHIKV remained to decrease by 2 dpsi, followed by a modest increase. In E/V-infected cells, no VEEV infectious particles were detectable even at 5 dpsi. Interestingly, E/V infection also completely prevented heterologous SINV replication by 4 dpsi. E/C did not inhibit SINV growth as efficiently as E/V but showed higher efficacy than EILV. The complete inhibition of SINV infection by E/V may partially result from the possibility that SINV and VEEV share the same receptors during infections (60).

## DISCUSSION

ISVs can alter insects' vector competence for arboviruses by either inducing superinfection exclusion or regulating vectors' immune systems (16, 61, 62). The first insect-specific alphavirus, EILV, was genetically characterized in 2012 and demonstrated to interfere with the growth of superinfecting alphaviruses (9, 59). However, its interaction with mosquitoes or mosquito cells remains inexplicit. Here, via transcriptome profiling of EILV-infected cells and RT-qPCR analysis of specific antiviral genes, we revealed that EILV infection mildly regulated host gene expression and antiviral response (Fig. 2–5). Phylogenetically, there is a strong relationship between EILV and several mosquito-borne alphaviruses, leading to a deduction that EILV lost its ability to invade vertebrate cells by an unknown mechanism

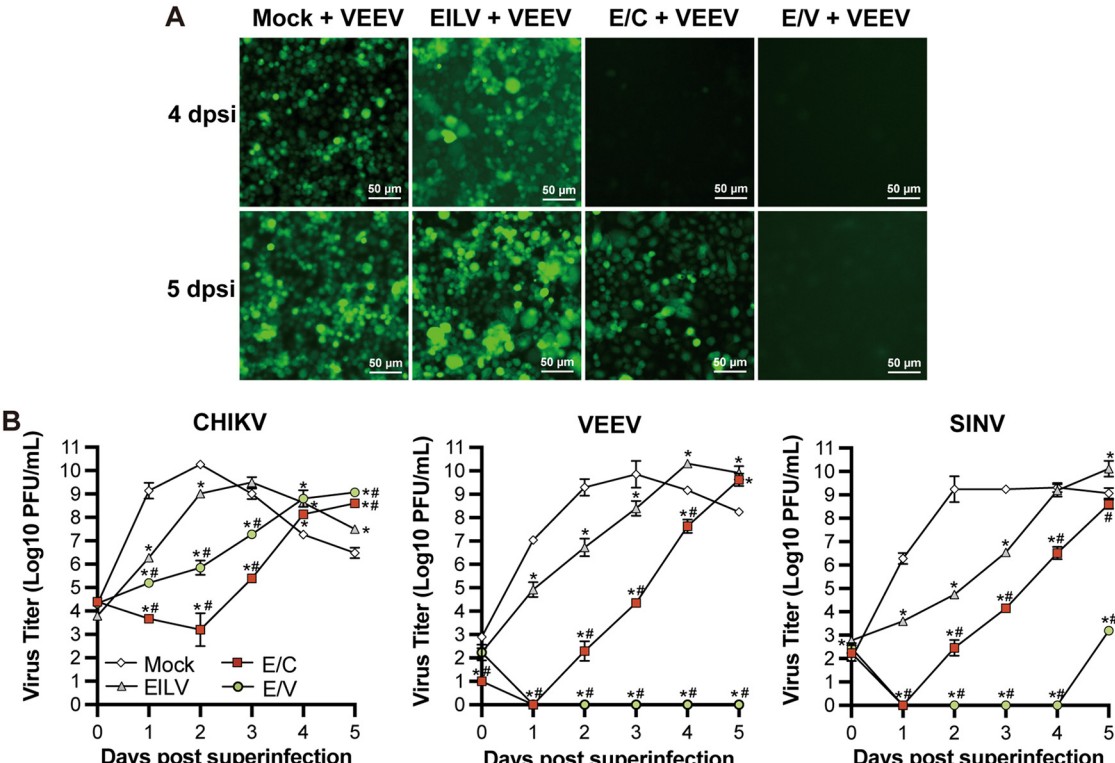

**FIG 7** EILV-based chimeras robustly inhibited pathogenic alphavirus growth in mosquito cells. (A) EILV- and chimera-induced VEEV growth inhibition determined via green fluorescent protein (GFP) expression. C6/36 cells were initially infected with EILV, E/C, or E/V at MOI 0.1. The superinfecting VEEV containing an extra subgenomic promoter expressing GFP (VEEV-GFP) was administered at the time point when the first virus reached the highest titer. Fluorescent photographs were taken at 4 and 5 days post-superinfection (dpsi). (B) EILV- and chimera-induced pathogenic alphavirus growth inhibition determined by growth kinetics. C6/36 cells were initially infected with EILV, E/C, or E/V at MOI 0.1. The superinfecting VEEV-GFP, CHIKV, and *Sindbis virus (SINV)* were administered at the time point when the first virus reached the highest titer. Cell supernatants were collected at intended time points, and titers were determined by plaque assay. Each sample was titrated twice ($n = 2$). Average titers $\pm$ SD are shown. * indicates differences that are statistically significant between virus- and mock-infected groups. # indicates differences that are statistically significant between EILV and chimeras.

(9). The weakened antiviral response induced by EILV infection in mosquito cells might result from an adaptive evolution, during which EILV acquires unique characteristics that facilitate its maintenance in mosquitoes. However, we also observed that the nonstructural genes of CHIKV and VEEV antagonized the antiviral effects raised by their structural genes (Fig. 6). Therefore, further investigation is needed to determine whether the limited antiviral defense following EILV infection stems from insect specificity of EILV or synergy of its nonstructural and structural genes.

Multiple EILV-based chimeras have been developed as vaccines and diagnostic platforms (12–14). Here, we found that these chimeras also hold tremendous potential to develop biological control measures against pathogenic alphaviruses (Fig. 7). Like parental EILV, these chimeras are restricted to infecting mosquitoes and therefore unharmful to humans and livestock. In contrast to the limited response elicited by EILV, they significantly potentiated antiviral defenses in host cells via different mechanisms (Fig. 2 to 5). Meanwhile, they contain the structural genes of pathogenic alphaviruses and are supposed to induce more robust superinfection exclusion than parental EILV. Combined with the ease of generating EILV-base chimeric alphaviruses, infection of mosquito populations with EILV-based chimeras can be tailored to reduce specific pathogenic alphavirus transmission in enzootic circulations.

We showed in the superinfection exclusion experiment that E/C and E/V indeed potently constrained the replication of CHIKV, VEEV, and SINV in C6/36 cells (Fig. 7). Although the initial infections were established at a low MOI (MOI 0.1), E/C and E/V significantly reduced superinfecting virus production. The differences in superinfecting virus titers between mock and chimera-infected cells can be as high as $10^7$-fold (E/C) and $10^9$-fold (E/V). Chimera

infections also delayed replication kinetics by 2 to 3 days during the 5-day monitoring period. In contrast, EILV infection inhibited superinfecting virus replication by 10 to $10^4$-fold and postponed growth kinetics by 1 to 2 days. Our data of EILV-mediated interference in C6/36 cells were consistent with a previous study where C7/10 cells were infected with EILV at MOI 10 to establish the initial infection (60). The previous study also provided *in vivo* evidence that the initial EILV infection of *Aedes aegypti* decreased CHIKV superinfection rate and delayed CHIKV dissemination from the midgut for 3 dpi (60). However, sustained interference was absent beyond 5 dpsi in mosquitoes. In the present study, the chimeras induced more significant interference than EILV *in vitro*, and the *in vivo* effects on mosquitoes should be evaluated in the future.

Our study also highlighted the intrinsic differences between the OW and NW alphaviruses. Alphaviruses are distributed throughout all continents, and the viruses circulating in the OW and NW evolved separately during the last thousands of years (63). As a result, they divergently modified their replication machinery and structural proteins for adaption in particular vectors and hosts. Variations in virus-host interactions between OW and NW viruses have been demonstrated in mammalian cells (63, 64), but little is known about their differences in virus-vector interplays in mosquito cells. Here, by utilizing previously constructed chimeric E/C and E/V, we examined virus-vector interaction discrepancies raised by CHIKV and VEEV structural proteins in infected mosquito cells. We observed that E/C efficiently promoted AMP production. Four out of six identified AMP products, including AttB, DEFA, DEFB, and DEFC, were significantly upregulated by E/C infection with fold changes greater than two (Fig. 3). E/V did not increase AMP transcription and even downregulated LYSC11 expression (Fig. 3). However, E/V infection robustly potentiated the transcription of the RNAi pathway components, including Dcr-2 in the siRNA pathway, AGO3 in the piRNA pathway, and Drosha, Dcr-1, and AGO1 in the miRNA pathway, while E/C infection exerted little effect on the RNAi pathways (Fig. 4). Additionally, although both E/C and E/V infections extensively modulated the apoptotic cascade, few apoptotic cells were observed postinfections (Fig. 5), suggesting a limited role played by apoptosis in antiviral defense.

We conducted the present study in C6/36 cells due to their prevalence in arbovirus research and the availability of a reference genome. Some limitations also came along. Using a cell line indisputably simplifies virus-vector interactions, and C6/36 cells are defective in the siRNA pathway, which is one of the most crucial antiviral mechanisms. Moreover, C6/36 genome is of high complexity. Most of the assembly contains duplicated sequences, and these duplications can complicate RNA-seq analysis by affecting gene count (65). Many genes were also not well annotated, and as a result, some DEGs are uncharacterized. However, we still featured a limited antiviral response elicited by an insect-specific alphavirus and demonstrated that its chimeric viruses could boost cellular defense. We also described the intrinsic discrepancies between the OW and NW alphaviruses from a new perspective. In-depth research utilizing different mosquito species would be of interest in the future.

## MATERIALS AND METHODS

**Viruses and cells.** The full-length cDNA clone of SINV strain Toto 1101 was kindly provided by Charles Rice's lab, the Rockefeller University, USA. The full-length cDNA clones of EILV isolate EO329 and CHIKV strain 181/25 were courtesy of Scott Weaver, University of Texas Medical Branch, Galveston, TX, USA. The recombinant EILV-based chimeric viruses were constructed and rescued as previously described (39). All the viruses were handled in the Biosafety Level 2 laboratory.

*Aedes albopictus* C6/36 cells and Baby Hamster Kidney BHK-21 cells were purchased from the American Type Culture Collection (ATCC), Manassas, VA, USA. C6/36 cells were maintained at 28℃ with 5% $CO_2$ in Dulbecco's Modified Eagle Medium (DMEM) containing 10% fetal bovine serum (FBS), 1% tryptose phosphate broth, 2 mM L-glutamine, 100 U/mL penicillin, and 100 $\mu$g/mL streptomycin. BHK-21 cells were cultivated at 37℃ with 5% $CO_2$ in Minimum Essential Medium $\alpha$ (MEM $\alpha$) supplemented with 10% FBS, vitamins, 2 mM L-glutamine, and penicillin-streptomycin.

**RNA-seq and data analysis.** C6/36 cells were infected with EILV, E/C, or E/V at MOI 0.1 or mock-infected in triplicates. Total RNA was isolated using TRIzol Reagent (Invitrogen, Waltham, MA, USA) when each virus reached the highest titer, that is, 1 dpi for E/C, 2 dpi for E/V, and 3 dpi for EILV. The total RNA of mock-infected cells was isolated at 1 dpi. Total RNA samples were sent to Groken Bioscience (Kowloon, HK, CHN) for transcriptome sequencing. Briefly, polyA+ transcripts were enriched and subjected to library preparation, and libraries were sequenced on a NovaSeq system to generate paired-end 150 bp reads.

Raw data were trimmed using Trimmomstic 0.39 (66), and resultant reads were aligned to viral genomes

and *Aedes albopictus* C6/36 genome (GenBank assembly accession: GCA_001876365.2) using TopHat 2.1.1 (67). Following quantification of the aligned transcripts via HTSeq 0.11.3 (68), DEGs between mock and virus-infected groups were obtained by edgeR 3.30.3 using the function glmQLFTest (69). FC > 2 and adjusted $P <$ 0.05 were taken as the threshold. Gene annotation was performed with eggNOG-mapper v2 with EggNOG v5.0 database and was listed in Table S3 (https://github.com/lrslab/Interplay-between-ISVs-and-mosquito-cells -suppl) (70, 71). GO analysis was conducted utilizing clusterProfiler 4.4.2. Adjusted $P <$ 0.01 were taken as the threshold.

**RT-qPCR analysis.** Total RNAs of mock or virus-infected cells were isolated as the RNA-seq section described. Subsequently, total RNA was reverse transcribed into first-strand cDNA using the High-Capacity cDNA Reverse transcription kit with RNase Inhibitor (Applied Biosystems, Waltham, MA, USA), followed by qPCR performed on an Applied Biosystems QuantStudio 7 using the Power SYBR green PCR Master Mix (Applied Biosystems). Data analysis was conducted based on the delta-delta Ct method. The 60S ribosomal protein L23 gene (RPL23) was selected for normalization. Primer sequences are listed in Table S5 (https:// github.com/lrslab/Interplay-between-ISVs-and-mosquito-cells-suppl).

**Apoptotic cell staining.** C6/36 cells seeded in 12-well plates ($2.5 \times 10^5$ cells/well) were infected at MOI 0.1 and fixed at 1, 2, and 3 dpi, respectively. Apoptotic cells were stained using the TUNEL BrightRed Apoptosis Detection kit (Vazyme, Nanjing, JS, CHN). Briefly, fixed cells were incubated in an equilibration buffer at room temperature for 30 min. End labeling was accomplished in a reaction mixture containing recombinant TdT enzyme and BrightRed labeling mix by incubation at 37℃ for 1 h. Following counterstaining using DAPI (Invitrogen), the fluorescence was visualized on a Nikon ECLIPSE Ti2 microscope.

**Superinfection exclusion.** C6/36 cells seeded in 6-well plates ($5 \times 10^5$ cells/well) were first infected with EILV, E/C, or E/V at MOI 0.1 or mock-infected. At the time point when each virus reached the highest ti-ter, that is, 1 dpi for E/C, 2 dpi for E/V, and 3 dpi for EILV, the cells were superinfected with VEEV-GFP, CHIKV, or SINV at MOI 0.1. Mock-infected cells were superinfected at 1 dpi. Replication of superinfecting viruses was monitored via fluorescence microscopy and growth kinetics. The fluorescence of GFP in VEEV-GFP-superin-fected cells was visualized on a Nikon ECLIPSE Ti2 microscope at 4 and 5 dpsi. Cell supernatants were col-lected at intended postsuperinfection time points and titrated on BHK-21 cells to plot the growth curves of superinfecting viruses.

**Statistical analysis.** Ordinary one-way ANOVA using Tukey's multiple-comparison test or ordinary two-way ANOVA using Tukey's multiple-comparison test was performed accordingly. All statistical calculations were performed in Prism 9 (GraphPad Software, San Diego, CA, USA).

## ACKNOWLEDGMENTS

This work was supported by the Hong Kong Branch of Southern Marine Science and Engineering Guangdong Laboratory (Guangzhou) (SMSEGL20SC02), Early career scheme (project number 9048204) from the Hong Kong Research Grant Council, Guangdong general research fund (project number 9240054) from Natural Science Foundation of Guangdong Province, and New Research Initiatives support from City University of Hong Kong (project number 9610497) to R.L. This work was also supported by the National Institutes of Health (R24AI120942) to Scott C. Weaver at the University of Texas Medical Branch, Galveston, Texas, and New Research Initiatives support from City University of Hong Kong (project number 9676013 and 9610451) to D.Y.K.

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
