## [Reviewer comments · Microbiology Spectrum]

Microbiology Spectrum

Insect-specific chimeric viruses potentiated antiviral responses and inhibited pathogenic alphavirus growth in mosquito cells

Lu Tan, Yiwen Zhang, Dal Young Kim, and Runsheng Li

Corresponding Author(s): Runsheng Li, City University of Hong Kong

Review Timeline:

Submission Date:	September 7, 2022
Editorial Decision:	October 17, 2022
Revision Received:	November 14, 2022
Accepted:	November 19, 2022

Editor: Bo Zhang

Reviewer(s): The reviewers have opted to remain anonymous.

Transaction Report:

DOI: <https://doi.org/10.1128/spectrum.03613-22>

October 17, 2022

Dr. Runsheng Li
City University of Hong Kong
Department of Infectious Diseases and Public Health
Room 2-502, 5/F, Block 2, To Yuen Building
31 To Yuen Street, City University of Hong Kong
Hong Kong
Hong Kong

Re: Spectrum03613-22 (Insect-specific chimeric viruses potentiated antiviral responses and curbed pathogenic alphavirus growth in the origin of mosquito cells)

Dear Dr. Runsheng Li:

The manuscript need to be polished thoroughly.

Link Not Available

Sincerely,

Bo Zhang

Journals Department
Reviewer comments:

Reviewer #1 (Comments for the Author):

The authors characterize the host response to infection with mosquito only alphavirus, Eilat virus, in C6/36 cell line. Utilizing chimeric viruses, they show that EILV is not able to a robust generate host response. In contrast, the two ELIV chimeras with structural ORFs representing Old (Chikungunya virus) and New World (Venezuelan equine encephalitis virus) viruses, elicit potent host responses. In addition, the responses generated utilize different pathways; E/C eliciting anti-microbial peptides, and E/V eliciting RNAi pathway. Furthermore, the authors perform superinfection exclusion experiments to demonstrate that infection

with EILV or the chimeras can profoundly inhibit subsequent infection with pathogenic alphaviruses. This research highlights the different evolutionary paths of dual host arbovirus and their likely host restricted ancestors such as Eilat virus. Few suggestions are stated below:

- 1) Superinfection exclusion experiments with EILV have been performed previously and warrant a discussion section.
- 2) The differences in host responses of the two *Aedes albopictus* cell lines, C7/10 and C6/36, should also be discussed.
- 3) Some additional editing grammar/style is needed throughout the manuscript. A few examples are below.
 - a. Sentences in lines 49-51 and 54-56 are run-on sentence, please consider breaking it into 2 sentences instead.
 - b. Sentences in lines 149-150 are not needed.
 - c. The word barely is used multiple times. Please consider rewording to "did not".
- 4) The figure legends are missing "limit of detection as indicated by the dashed lines".

Reviewer #2 (Comments for the Author):

The manuscript "Insect-specific chimeric viruses potentiated antiviral responses and curbed pathogenic alphavirus growth in the origin of mosquito cells" by L. Tan et al. describes interactions of alphaviruses and their insect-specific chimeras with insect cells, C6/36 in particular. They describe various changes in cell response to viral infection by insect-specific alphavirus Eilat, and its two chimeras, EILV/CHIKV and EILV/VEEV where genes encoding structural proteins of Eilat virus were replaced by corresponding genes from CHIKV and VEEV. The authors also analyze various antiviral cell responses in reaction to EILV/CHIKV, EILV/VEEV, and Eilat viruses; and to superinfection by CHIKV, SINV, and VEEV.

Comments:

Not clear why is it "curbed" alphavirus growth and why is it "in the origin of mosquito cells". May be the title needs to be edited. In the previous publication of the authors (reference 39 in the current manuscript, "A Productive Expression Platform Derived from Host-Restricted Eilat Virus: Its Extensive Validation and Novel Strategy", *Viruses*) they explained large drop in virus titers for the chimeras after peaked values by decreased stability of the chimeras. That would be the simplest explanation, but in the current manuscript they rationalize the same phenomenon differently, by increased antiviral cell response. It would be good to include discussion of the differences between the previous and the current explanations with the reasoning why they changed their mind in explaining the phenomenon.

When observed insignificant changes, it would be suitable to just say so, instead of writing long description of detected variations in cell responses to viral infection. It is of course essential to summarize observed effects and to discuss their importance. That would make the manuscript more concise. For example, in their "Viral infections modulated cell death pathways" section their conclusion was that the changes they saw were not significantly different from mock infection, they could shorten that part of the manuscript simply stating that pathway did not seem to be significantly affected by virus infection. The authors do not specify which VEEV strain was used, if it is the wild type, they need to specify the biosafety level and safety precautions taken

The manuscript would greatly benefit from English improvement. It is quite difficult to read, with some typos ("GEP" instead "GFP", or "overserved" most likely meaning "observed"), and cumbersome phraseology present.

Some statements should be corrected, e.g. the authors state that Alphavirus is the single genus in *Togaviridae* family. In fact, the family contains two genera, Alphavirus, and Rubivirus. Other statements are questionable, like "VEEV is the most important human and equine New World alphavirus", there are others like EEEV, WEEV, etc. that are important as well.

Some phrases are strangely worded, for example, "1395 DEGs were peculiarly modulated"; "viruses... could be developed as a safe and intriguing tool"; "host apoptotic cascade was extensively and controversially regulated", etc.

They love word "fold" but use it in many cases incorrectly. "Fold change" could mean "protein or nucleic acid structure or conformation change" meaning that amino-acid sequence is differently spatially folded, or could have other meanings as well. But the authors mean in the manuscript "multiplication factor". And "increased by 0.72-fold and 0.26-fold" would actually mean decrease in quantity, not an increase. That term "fold" has to be changed throughout the manuscript.

Expressions like "something was also regulated" do not tell reader much, the authors need to be more specific, were these factors/genes, etc. up- or down regulated? Or just changed?

Figure 1C: wrong abscissa axis, label 6 should be moved farther from label 3

Figure 3B: I am not sure asterisks are necessary, standard deviation bars should be enough for reader to judge significance of changes observed relative to mock infection. Same for other Figures.

Staff Comments:

Preparing Revision Guidelines

To submit your modified manuscript, log onto the eJP submission site at <https://spectrum.msubmit.net/cgi-bin/main.plex>. Go to Author Tasks and click the appropriate manuscript title to begin the revision process. The information that you entered when you

first submitted the paper will be displayed. Please update the information as necessary. Here are a few examples of required updates that authors must address:

Please return the manuscript within 60 days; if you cannot complete the modification within this time period, please contact me. If you do not wish to modify the manuscript and prefer to submit it to another journal, please notify me of your decision immediately so that the manuscript may be formally withdrawn from consideration by Microbiology Spectrum.

Reviewer comments:

Reviewer #1 (Comments for the Author):

The authors characterize the host response to infection with mosquito only alphavirus, Eilat virus, in C6/36 cell line. Utilizing chimeric viruses, they show that EILV is not able to robustly generate host response. In contrast, the two ELIV chimeras with structural ORFs representing Old (Chikungunya virus) and New World (Venezuelan equine encephalitis virus) viruses, elicit potent host responses. In addition, the responses generated utilize different pathways; E/C eliciting anti-microbial peptides, and E/V eliciting RNAi pathway. Furthermore, the authors perform superinfection exclusion experiments to demonstrate that infection with EILV or the chimeras can profoundly inhibit subsequent infection with pathogenic alphaviruses. This research highlights the different evolutionary paths of dual host arbovirus and their likely host restricted ancestors such as Eilat virus. Few suggestions are stated below:

1) Superinfection exclusion experiments with EILV have been performed previously and warrant a discussion section.

Many thanks for the suggestion. We added a paragraph in the discussion section to highlight the superior interference induced by the chimeras and compare the superinfection exclusion result with the previously published data. Below is the paragraph content:

Line 288-301: “We showed in the superinfection exclusion experiment that E/C and E/V indeed potently constrained the replication of CHIKV, VEEV, and SINV in C6/36 cells (Fig. 7). Although the initial infections were established at a low MOI (MOI 0.1), E/C and E/V significantly reduced superinfecting virus production. The differences in superinfecting virus titers between mock and chimera-infected cells can be as high as 10^7 -fold (E/C) and 10^9 -fold (E/V). Chimera infections also delayed replication kinetics by 2-3 days during the 5-day monitoring period. In contrast, EILV infection inhibited superinfecting virus replication by 10 to 10^4 -fold and postponed growth kinetics by 1-2 days. Our data of EILV-mediated interference in C6/36 cells were consistent with a previous study where C7/10 cells were infected with EILV at MOI 10 to establish the initial infection (60). The previous study also provided *in vivo* evidence that the initial EILV infection of *Aedes aegypti* decreased CHIKV superinfection rate and delayed CHIKV dissemination from the midgut for 3 dpi (60). However, sustained interference was absent beyond 5 dpi in mosquitoes. In the present study, the chimeras induced more significant interference than EILV *in vitro*, and the *in vivo* effects on mosquitoes should be evaluated in the future.”

2) The differences in host responses of the two *Aedes albopictus* cell lines, C7/10 and C6/36, should also be discussed.

Many thanks for the comment. In the present manuscript, we did not raise the differences in host responses between C7/10 and C6/36 cells for a few reasons:

In our experience, C7/10 and C6/36 cells showed mild differences against EILV infection. According to our data and previously published data (Nasar et al., 2015; Tan et al., 2021), the growth curves of EILV and its chimeras were similar in C7/10 and C6/36 cells, and the interference induced by EILV was also alike in both cell lines. As C7/10 and C6/36 are of the same origin, they are fundamentally similar. We suppose the differences between these two cell lines have a minor effect on our findings.

Secondly, although C7/10 and C6/36 cells are both used to grow arboviruses, the studies investigating virus-vector interactions exclusively adopted C6/36 cells; instead, C7/10 cell line is a useful model for understanding hormone responses in insects (Walker et al., 2014). As a result, little

has been reported about C7/10 cell antiviral responses. Comprehensive comparisons between C7/10 and C6/36 require further studies.

3) Some additional editing grammar/style is needed throughout the manuscript. A few examples are below.

Many thanks for the suggestion. We corrected several grammatic mistakes and modified the wording of our manuscript thoroughly. We avoided run-on sentences and repeated words, aiming to make the manuscript more readable. All the changes are highlighted in the revised manuscript.

a. Sentences in lines 49-51 and 54-56 are run-on sentence, please consider breaking it into 2 sentences instead.

We modified these sentences accordingly:

Lines 49-51 "*Alphavirus*, the sole genus in the *Togaviridae* family, is a group of enveloped viruses encompassing a positive-strand genomic RNA (1). They are classified as the Old World (OW) and New World (NW) types according to where they circulate (2)."

Lines 54-56 "Chikungunya virus (CHIKV), a typical OW alphavirus, can cause fever and debilitating joint pain despite low fatality. It has brought several epidemic outbreaks since 2004 (3)."

b. Sentences in lines 149-150 are not needed.

We deleted the corresponding sentences.

c. The word barely is used multiple times. Please consider rewording to "did not".

We reworded "barely" in the following sentences:

Line 24 "In contrast, EILV triggered a limited antiviral response."

Line 156-157 "In contrast to chimera infections, EILV infection modulated few of these signaling components (Table 1)."

4) The figure legends are missing "limit of detection as indicated by the dashed lines".

Many thanks for the reminder. The dashed lines in Fig. 3-5 indicate a fold change value of 1, which corresponds to an unchanged gene expression level. The values above the dashed line represent upregulation, while the values below the line mean downregulation. We added this explanation in the figure legends.

Reviewer #2 (Comments for the Author):

The manuscript "Insect-specific chimeric viruses potentiated antiviral responses and curbed pathogenic alphavirus growth in the origin of mosquito cells" by L. Tan et al. describes interactions of alphaviruses and their insect-specific chimeras with insect cells, C6/36 in particular. They describe various changes in cell response to viral infection by insect-specific alphavirus Eilat, and its two chimeras, EILV/CHIKV and EILV/VEEV where genes encoding structural proteins of Eilat virus were replaced by corresponding genes from CHIKV and VEEV. The authors also analyze various antiviral

cell responses in reaction to EILV/CHIKV, EILV/VEEV, and Eilat viruses; and to superinfection by CHIKV, SINV, and VEEV.

Comments:

Not clear why is it "curbed" alphavirus growth and why is it "in the origin of mosquito cells". May be the title needs to be edited.

Many thanks for the comment. We reworded the title to make it more concise and understandable. Here is the new title: "Insect-specific chimeric viruses potentiated antiviral responses and inhibited pathogenic alphavirus growth in mosquito cells."

In the previous publication of the authors (reference 39 in the current manuscript, "A Productive Expression Platform Derived from Host-Restricted Eilat Virus: Its Extensive Validation and Novel Strategy", *Viruses*) they explained large drop in virus titers for the chimeras after peaked values by decreased stability of the chimeras. That would be the simplest explanation, but in the current manuscript they rationalize the same phenomenon differently, by increased antiviral cell response. It would be good to include discussion of the differences between the previous and the current explanations with the reasoning why they changed their mind in explaining the phenomenon.

Many thanks for the comment. In our previous publication, we performed a comparative analysis of the replication of EILV-based chimeras and noticed large drops in virus titers after peaking. We then declared that the chimeras displayed inferior stabilities compared with the parental virus. However, this can be one of the reasons why the chimera titers dropped quickly. Other infection phenotypes, including plaque morphology and cytopathic effect, strongly suggested a difference in virus-host interactions, which might also contribute to the titer drop.

In the present manuscript, we described all these infection phenotypes. However, we emphasized the dramatic titer drop of the chimeras as an indication of increased antiviral responses. We overlooked the changes in plaque morphology and cytopathic effect. We realized that our statement in the present manuscript was inaccurate. Therefore, we modified this part in the Introduction and Result sections as follows:

Line 92-96: "In a previous study, we constructed a group of EILV-based chimeras containing the structural genes derived from seven different OW and NW pathogenic alphaviruses. These chimeras displayed infection phenotypes distinct from parental EILV in *Aedes albopictus* C7/10 cells (39), suggesting potential divergences in virus-vector interactions."

Line 110-120: "Here, we further described E/C's and E/V's infection phenotypes in *Aedes albopictus* C6/36 cells, one of the most widely used insect cell lines. E/C and E/V were generated by replacing the structural genes of EILV with that of CHIKV strain Caribbean or VEEV strain TC83 (Fig. 1A). Like in C7/10 cells, the chimeras formed clearer plaques than parental EILV on C6/36 monolayer (Fig. 1A). They also induced apparent cytopathic effect (CPE) and inhibited host cell multiplication, while EILV had little impact on cell growth (Fig. 1B). Consistent with the observation in C7/10 cells, the chimeras' growth curves dropped dramatically after peaking (Fig. 1C). At 6 dpi, E/C's titer showed a 268-fold decrease relative to its peak value, and E/V's titer displayed a 98-fold decrease. Instead, a modest 12-fold drop was observed in the case of EILV. These differences in infection phenotype between EILV and its chimeras suggested their variations in virus-vector interplays."

When observed insignificant changes, it would be suitable to just say so, instead of writing long description of detected variations in cell responses to viral infection. It is of course essential to summarize observed effects and to discuss their importance. That would make the manuscript more concise. For example, in their "Viral infections modulated cell death pathways" section their

conclusion was that the changes they saw were not significantly different from mock infection, they could shorten that part of the manuscript simply stating that pathway did not seem to be significantly affected by virus infection.

Many thanks for the comment. We have shortened the descriptions for non-significant results. For the "Viral infections modulated cell death pathways" section, although the TUNEL showed little difference between the viruses and the mock infection, it indicated the effect of simultaneous pro- and anti-apoptotic regulation. We described these findings to emphasize that the cell death pathways were comprehensively regulated by viral infections, especially by E/V infection. The revised section is shown as follows:

Line 204-220: "Many cells undergo apoptosis upon infections, which can be a crucial mechanism deployed by the host to curtail virus propagation and dissemination (55). Here a considerable number of DEGs induced by viral infections were identified to regulate apoptotic signaling and functioning in both pro- and anti-apoptotic ways (Fig. 2D). The core apoptotic pathway in mosquitoes largely resembles that of *Drosophila melanogaster* (56, 57) (Fig. 5A). Following infections, transcriptions of several inhibitor-of-apoptosis (IAP) genes, including IAP1, IAP2, and IAP6, were significantly potentiated, and E/V posed the most substantial effect (Fig. 5B). Meanwhile, viral infections also raised the expression levels of initiator caspase Dronc and a few effector caspases (Fig. 5A and C); fold changes triggered by E/V infection were also the greatest. To identify the apoptotic status of infected cells, terminal deoxynucleotidyl transferase dUTP nick end labeling (TUNEL) assay was performed to detect DNA fragmentation (Fig. 5D). Compared to mock-infected cells, viral infections slightly increased the number of apoptotic cells, with no significant difference between EILV and chimera infections. These results suggest that apoptosis plays a minor role in virus-vector interactions due to simultaneous pro- and anti-apoptotic regulations. Autophagy was also affected by viral infections (Fig. 2D; Table S4). However, the function of autophagy in mosquitoes during arbovirus infection is understudied (58), and a limited role of autophagy has been reported (59)."

The authors do not specify which VEEV strain was used, if it is the wild type, they need to specify the biosafety level and safety precautions taken

Many thanks for the reminder. We added strain or isolate information for all the viruses used in the present study. Specifically, we used the vaccine strain TC83 of VEEV. All the viruses were handled in the BSL-2 lab. We also added this information in the Method section:

Line 338-339: "All the viruses were handled in the Biosafety Level 2 laboratory."

The manuscript would greatly benefit from English improvement. It is quite difficult to read, with some typos ("GEP" instead "GFP", or "overserved" most likely meaning "observed"), and cumbersome phraseology present.

Many thanks for the corrections. We checked through the manuscript and corrected the typos we found. We also replaced the cumbersome phrases with simpler expressions. All the changes are highlighted in the revised manuscript.

Some statements should be corrected, e.g. the authors state that Alphavirus is the single genus in Togaviridae family. In fact, the family contains two genera, Alphavirus, and Rubivirus. Other statements are questionable, like "VEEV is the most important human and equine New World alphavirus", there are others like EEEV, WEEV, etc. that are important as well.

Many thanks for the suggestions. Regarding the first question, *Togaviridae* contained the genus *Rubivirus* before April 2019. According to ICTV, *Rubivirus* has now been moved to the family

Matonaviridae (<https://ictv.global/report/chapter/togaviridae/togaviridae>). Therefore, we did not change this sentence. For the questionable statement saying "VEEV is the most important human and equine New World alphavirus", we revised it into "VEEV is one of the essential NW alphaviruses affecting humans and equines".

Some phrases are strangely worded, for example, "1395 DEGs were peculiarly modulated"; "viruses... could be developed as a safe and intriguing tool"; "host apoptotic cascade was extensively and controversially regulated", etc.

Many thanks for the advice. We modified the wording of our manuscript to make it more readable.

Some modifications are indicated as follows:

Line 134-135: "1639 DEGs were specific for E/C infection, and 1395 DEGs were specific for E/V infection"

Line 76-77: "Taken together, mosquito-only alphaviruses can be a useful tool to modulate vector competence and contain pathogenic alphavirus transmission."

Line 204-206: "Here a considerable number of DEGs induced by viral infections were identified to regulate apoptotic signaling and functioning in both pro- and anti-apoptotic ways."

They love word "fold" but use it in many cases incorrectly. "Fold change" could mean "protein or nucleic acid structure or conformation change" meaning that amino-acid sequence is differently spatially folded, or could have other meanings as well. But the authors mean in the manuscript "multiplication factor". And "increased by 0.72-fold and 0.26-fold" would actually mean decrease in quantity, not an increase. That term "fold" has to be changed throughout the manuscript.

Many thanks for the comment. In bioinformatics, "fold change" is often used in analyzing gene expression data for measuring changes in the expression level of a gene. It is defined as the ratio between an original and subsequent measurement; for original gene expression level A and subsequent gene expression level B , the fold change of B with respect to A is B/A . Moreover, according to the Oxford English Dictionary and Merriam-Webster Dictionary, "-fold" is synonymous with "times", meaning multiplied by a specified number.

In the original manuscript, we meant a 1.72-fold change and a 1.26-fold change by saying "increased by 0.72-fold and 0.26-fold". This indeed caused some confusion and ambiguity. Therefore, we changed all the expressions like "increased by 0.72-fold and 0.26-fold" to "increased to 1.72-fold and 1.26-fold". All the changes are highlighted in the revised manuscript.

Expressions like "something was also regulated" do not tell reader much, the authors need to be more specific, were these factors/genes, etc. up- or down regulated? Or just changed?

Many thanks for the suggestion. We added some details to make our expressions more informative. Below is one example:

Line 130-133: "Of EILV-induced DEGs, 659 genes (43%) were similarly modulated during E/C infection, with 504 upregulated and 155 downregulated. A total of 1089 communal DEGs (72% of EILV-induced DEGs) were detected with EILV and E/V infections, of which 761 were upregulated and 328 were downregulated."

Figure 1C: wrong abscissa axis, label 6 should be moved farther from label 3

Many thanks for the suggestion. We added a gap in the abscissa axis between label 3 and 6 to indicate the distance. The new figure is shown as follows:

Figure 3B: I am not sure asterisks are necessary, standard deviation bars should be enough for reader to judge significance of changes observed relative to mock infection. Same for other Figures.

Many thanks for the comment. The expression levels of most genes changed a lot, and standard deviation bars are indeed sufficient to judge significance. However, in a few cases, e.g. Fig. 4B, some gene expression levels did not change dramatically, but these changes were determined to be statistically significant. Therefore, it would be better to add asterisks to tell the readers if the changes are statistically significant.

November 19, 2022

Dr. Runsheng Li
City University of Hong Kong
Department of Infectious Diseases and Public Health
Room 2-502, 5/F, Block 2, To Yuen Building
31 To Yuen Street, City University of Hong Kong
Hong Kong
Hong Kong

Re: Spectrum03613-22R1 (Insect-specific chimeric viruses potentiated antiviral responses and inhibited pathogenic alphavirus growth in mosquito cells)

Dear Dr. Runsheng Li:

Your manuscript has been accepted, and I am forwarding it to the ASM Journals Department for publication. You will be notified when your proofs are ready to be viewed.

Sincerely,

Bo Zhang
Editor, Microbiology Spectrum

Journals Department
Supplemental file 2: Accept
Supplemental file 1: Accept